# Hybrid Integrated Silicon Photonics Based on Nanomaterials

Domenic Prete [1,*], Francesco Amanti [2], Greta Andrini [3,4], Fabrizio Armani [5], Vittorio Bellani [1,6], Vincenzo Bonaiuto [7,8], Simone Cammarata [2,9], Matteo Campostrini [10], Samuele Cornia [1,11], Thu Ha Dao [7,8], Fabio De Matteis [7,8], Valeria Demontis [1,12], Giovanni Di Giuseppe [13,14], Sviatoslav Ditalia Tchernij [4,15], Simone Donati [2,16], Andrea Fontana [1], Jacopo Forneris [4,15], Roberto Francini [7,8], Luca Frontini [5,17], Gian Carlo Gazzadi [18], Roberto Gunnella [13,14], Simone Iadanza [19], Ali Emre Kaplan [20], Cosimo Lacava [1,21], Valentino Liberali [5,17], Leonardo Martini [11], Francesco Marzioni [13,14,22], Claudia Menozzi [11], Elena Nieto Hernández [4,15], Elena Pedreschi [2], Paolo Piergentili [13,14], Paolo Prosposito [7,8], Valentino Rigato [10], Carlo Roncolato [10], Francesco Rossella [1,11], Andrea Salamon [8], Matteo Salvato [23], Fausto Sargeni [8,24], Jafar Shojaii [25], Franco Spinella [2], Alberto Stabile [5,17], Alessandra Toncelli [2,16], Gabriella Trucco [5,26] and Valerio Vitali [1,21]

1       Istituto Nazionale di Fisica Nucleare, Sezione di Pavia, Via Agostino Bassi 6, 27100 Pavia, Italy; vittorio.bellani@unipv.it (V.B.); valerio.vitali@unipv.it (V.V.)
2       Istituto Nazionale di Fisica Nucleare, Sezione di Pisa, 56127 Pisa, Italy; franco.spinella@pi.infn.it (F.S.); alessandra.toncelli@unipi.it (A.T.)
3       Dipartimento di Elettronica e Telecomunicazioni, Politecnico di Torino, 10129 Torino, Italy; greta.andrini@to.infn.it
4       Istituto Nazionale di Fisica Nucleare, Sezione di Torino, 10125 Torino, Italy
5       Istituto Nazionale di Fisica Nucleare, Sezione di Milano, 20133 Milano, Italy; luca.frontini@mi.infn.it
6       Dipartimento di Fisica, Università di Pavia, 27100 Pavia, Italy
7       Dipartimento di Ingegneria Industriale, Università di Roma Tor Vergata, 00133 Roma, Italy; vincenzo.bonaiuto@uniroma2.it (V.B.); fabio.dematteis@roma2.infn.it (F.D.M.); francini@roma2.infn.it (R.F.)
8       Istituto Nazionale di Fisica Nucleare, Sezione di Roma Tor Vergata, 00133 Roma, Italy; andrea.salamon@roma2.infn.it
9       Dipartimento di Ingegneria dell'Informazione, Università di Pisa, 56122 Pisa, Italy
10      Laboratori Nazionali di Legnaro, Istituto Nazionale di Fisica Nucleare, 35020 Legnaro, Italy; matteo.campostrini@lnl.infn.it (M.C.); valentino.rigato@lnl.infn.it (V.R.)
11      Dipartimento di Scienze Fisiche, Informatiche e Matematiche, Università di Modena e Reggio Emilia, 41125 Modena, Italy
12      National Enterprise for nanoScience and nanoTechnology, Scuola Normale Superiore, Istituto Nanoscienze—Consiglio Nazionale delle Ricerche, 56127 Pisa, Italy
13      Scuola di Scienze e Tecnologie, Divisione di Fisica, Università di Camerino, 62032 Camerino, Italy; francesco.marzioni@unicam.it (F.M.); paolo.piergentili@unicam.it (P.P.)
14      Istituto Nazionale di Fisica Nucleare, Sezione di Perugia, 06123 Perugia, Italy
15      Dipartimento di Fisica, Università di Torino, 10125 Torino, Italy
16      Dipartimento di Fisica, Università di Pisa, 56127 Pisa, Italy
17      Dipartimento di Fisica, Università di Milano, 20133 Milano, Italy
18      Istituto Nanoscienze—Centro S3, Consiglio Nazionale delle Ricerche, Via Campi 213/A, 41125 Modena, Italy; giancarlo.gazzadi@nano.cnr.it
19      Paul Scherrer Institute, 5232 Villigen, Switzerland; simone.iadanza@psi.ch
20      Optoelectronics Research Center, University of Southampton, Southampton SO17 1BJ, UK
21      Dipartimento di Ingegneria Industriale e dell'Informazione, Università di Pavia, 27100 Pavia, Italy
22      Dipartimento di Fisica, Università di Napoli "Federico II", 80126 Napoli, Italy
23      Dipartimento di Fisica, Università di Roma Tor Vergata, 00133 Roma, Italy
24      Dipartimento di Ingegneria Elettronica, Università di Roma Tor Vergata, 00133 Roma, Italy
25      Space Technology and Industry Institute, Swinburne University of Technology, Hawthorn, VIC 3122, Australia; jshojaii@swin.edu.au
26      Dipartimento di Informatica, Università di Milano, 20133 Milano, Italy
*       Correspondence: domenic.prete@pv.infn.it

**Abstract:** Integrated photonic platforms have rapidly emerged as highly promising and extensively investigated systems for advancing classical and quantum information technologies, since their ability to seamlessly integrate photonic components within the telecommunication band with existing silicon-based industrial processes offers significant advantages. However, despite this integration

facilitating the development of novel devices, fostering fast and reliable communication protocols and the manipulation of quantum information, traditional integrated silicon photonics faces inherent physical limitations that necessitate a challenging trade-off between device efficiency and spatial footprint. To address this issue, researchers are focusing on the integration of nanoscale materials into photonic platforms, offering a novel approach to enhance device performance while reducing spatial requirements. These developments are of paramount importance in both classical and quantum information technologies, potentially revolutionizing the industry. In this review, we explore the latest endeavors in hybrid photonic platforms leveraging the combination of integrated silicon photonic platforms and nanoscale materials, allowing for the unlocking of increased device efficiency and compact form factors. Finally, we provide insights into future developments and the evolving landscape of hybrid integrated photonic nanomaterial platforms.

**Keywords:** integrated silicon photonics; nanostructured materials; hybrid photonic platforms; integrated photonic circuits

## 1. Introduction

The development of modern electronics has boosted the generation of unprecedented amounts of data, which are becoming progressively more complex to store and, most importantly, to transfer from server to client [1,2]. In this scenario, photonic technologies offered a relatively easy and very effective way to transfer information in little time and over wide distances [3–5]. Nonetheless, among the several contenders in the field, over the past years silicon photonics has emerged as an excellent platform for developing devices capable of manipulating and transferring optical signals, succeeding in offering high-performance components for telecommunications [6,7]. Building upon the fundamental building block of silicon (Si) waveguides [8,9], several optical components can be realized, including—but not limited to—Bragg gratings [10], directional couplers [11] and arrayed waveguide gratings [12,13]. Silicon photonic devices modulating optical signals have also been demonstrated, such as ring resonators [14], high-speed optical modulators [15–17] and Mach–Zehnder interferometers [18–20]. Figure 1 reports some relevant developments achieved by silicon photonics for both classical and quantum information and communication technologies. These components have served as valuable demonstrators and enablers of a technology that is increasingly posing itself as the standard in information and communication technologies for the time to come.

Indeed, integrated silicon photonic platforms, which leverage their robust expertise in device fabrication through the CMOS technological platform [21,22], have not only proven their validity in providing high-efficiency photonics devices able to route optical signals over complex geometries with the realization of bend multi-mode waveguides [23], or over long distances with low losses [24], but also in being an effective platform to boost quantum technologies [25–33]. Indeed, quantum information can be encoded in photons propagating in multiple paths fabricated in silicon waveguides, and operations can be performed on by employing, for instance, directional couplers [34,35], opening the path to the realization of quantum processors [36]. Furthermore, silicon photonics is identified to be the enabling technology for the development of Quantum Key Distribution, allowing for secure communication protocols for quantum information over long distances [37,38]—as discussed in several works [25,39]—while being compatible with wavelengths used in standard telecommunication protocols [40].

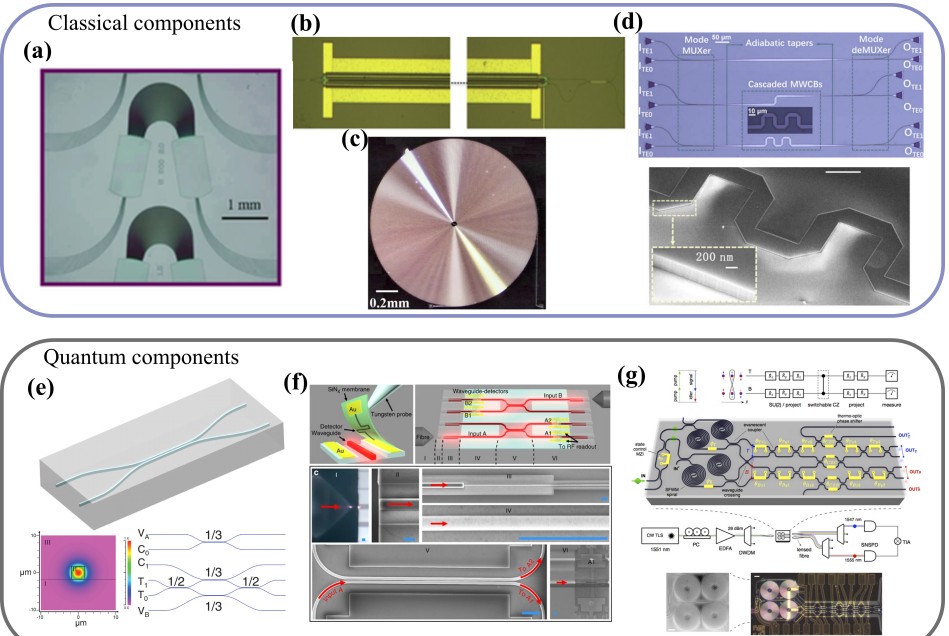

**Figure 1.** Classical and quantum optical components developed within the framework of silicon photonics. (**a**) Optical micrograph of an arrayed waveguide grating for optical multiplexing and demultiplexing. Image reprinted from [13], copyright Wiley 2013. (**b**) Optical micrograph of a Mach−Zehnder Modulator. Image reprinted from [41]. (**c**) Optical micrograph of a 50 cm spiral low−loss waveguide. Image reprinted from [24]. (**d**) Optical micrograph of a photonic integrated circuit employing Multimode Waveguide Corner Bend (MWCB) structures for signal routing. A scanning electron micrograph of an MWCB structure is reported in the lower panel. Image reprinted from [23]. (**e**) Building blocks for quantum photonic platforms integrated in silicon, namely, integrated waveguides. The upper panel reports a pictorial view and the lower−left panel reports the computed intensity profile of a guided mode. The lower−right panel reports the schematic for the implementation of a CNOT circuit using guided modes in a silicon photonic integrated circuit. Image reprinted from [34], copyright AAAS 2008. (**f**) Integration of single−photon nanowire detectors in photonic integrated circuits. Image reprinted from [42]. (**g**) Pictorial view and optical micrograph of a photonic circuit implementing two−qubit entanglement. Image reprinted from [36].

Nonetheless, despite its numerous advantages and promising features, silicon photonics is still trying to mitigate some limitations imposed by the material platform and technology [43,44], namely, the large physical footprint of integrated photonic devices [45] and the need for integrated light sources and detectors [46,47]. The former is caused by the dimensions of some photonic components which constrain the footprint for Photonic Integrated Circuits (PICs) and makes their integration in current microelectronic platforms a challenge [48]. On the other hand, integrating optical sources and detectors on-chip with silicon-only technology is limiting, due to the physical properties of the material—i.e., indirect bandgap—hampering its effectiveness as when realizing photoemitters and photodetectors [49].

To solve these issues, and to increase device performance, the integration of nanotechnology and nanostructures in PICs is taking place. Indeed, nanotechnology allows for the precise fabrication of components, enabling the control and manipulation of optical fields at the sub-micrometer scale [50], as well as the creation of compact and highly efficient devices such as modulators, detectors, light sources and polarization rotators [51]. This comes from the exploitation of the favorable optical properties of purposefully engineered nanostructures, which have been found to have several promising applications such as biosensing [52,53], sustainable energy generation [54] and photon quantum light emitting [55,56], to name a few. In this scenario, the combination of silicon photonic integrated platforms with nanostructured materials aims to obtain the advantage of both worlds.

Indeed, semiconducting nanostructures can improve relevant photonic device parameters, e.g., by reducing spatial footprint and allowing the realization of compact photonic devices, or by allowing the integration of on-chip light detectors with high responsivity and low dark currents. Furthermore, nanostructures integrated in silicon photonic platforms to implement light modulation feature high modulation efficiency or enable polarization control with reduced losses. Ultimately, nanostructures integration in silicon photonic platforms provides access to the realization of scalable devices featuring high performance and novel functionalities without compromising overall size [11,57].

Some excellent works in the literature report on the main results achieved by the fruitful combination of nanostructures and silicon photonics, often focusing on a single material platform [51] or on specific applications [58,59]. This review aims to illustrate the main techniques developed by researchers to integrate semiconducting nanostructures in silicon photonic platforms, as well as some of the main related advancements. The integration of nanostructures into silicon photonic devices is comprehensively examined, describing the main methods developed for the realization of photonic devices combining integrated silicon platforms and nanomaterials, elucidating key factors that contribute to heightened performance and a minimized spatial footprint. The primary objective of this review is to provide interested readers with an in-depth understanding of the integration of nanomaterials onto photonic platforms. By doing so, it aims to provide insights into the evolution of hybrid photonic platforms that seamlessly combine silicon technologies with cutting-edge nanostructure fabrication techniques.

Section 2.1 will provide a brief description of each nanomaterial class, giving insights on the main properties of each material. It will further describe several techniques to integrate such nanomaterials in photonic devices and waveguides, highlighting the main advantages and disadvantages of individual methods.

Section 2.2 will focus on the main results achieved by the integration of 1D nanostructures, e.g., semiconducting nanowires, carbon nanotubes and graphene nanoribbons, into silicon photonic platforms.

Section 2.3 will focus on the progress in devices incorporating 2D materials such as graphene and Transition Metal Dichalcogenides (TMDs) on silicon waveguides.

## 2. Hybrid Integrated Photonic Platforms

### 2.1. Semiconducting Nanostructures and Integration Techniques

Figure 2 reports the main classes of semiconducting nanostructures generally available in nanotechnological platforms, classified in 1D and 2D, as well as further details regarding specific nanostructures which are commonly employed in combination with photonic integrated circuits.

Among one-dimensional nanostructures, nanowires stand out due to their high aspect ratio, unprecedented crystal quality and access to complex heterostructures owing to strain relaxation [60]. These features has led this nanomaterial class to being at the foundation of several advances for a plethora of applications, including energy storage and harvesting [61], optoelectronics [62] and quantum technologies [63]. Furthermore, nanowires hosting quantum dots are a key technology for the realization of single photon emitters [64] and detectors [65]. Nanoribbons, with their flattened geometry, feature favorable electrical transport properties [66] and advanced control of response to optical stimuli [67]. Carbon nanotubes (CNTs), with their exceptional electrical and thermal properties, offer exciting possibilities for high-performance photonic devices [68]. The diverse electronic properties based on their chirality allow for both semiconducting and metallic behavior [69], broadening their potential applications [70].

Moving towards two-dimensional nanostructures, graphene possesses exceptional electrical transport properties, e.g., high carrier mobility [71]. Its properties show promise for light modulators [72] and photodetectors [73]. However, its inherent lack of a bandgap limits its use in applications like light-emitting devices. In contrast, transition metal dichalcogenides (TMDs) feature diverse properties depending on their composition and

structure, offer strong light–matter interaction and tunable bandgaps [74]. This makes them suitable for a wide range of photonic devices like light emitters [75], photodetectors [76] and modulators [77].

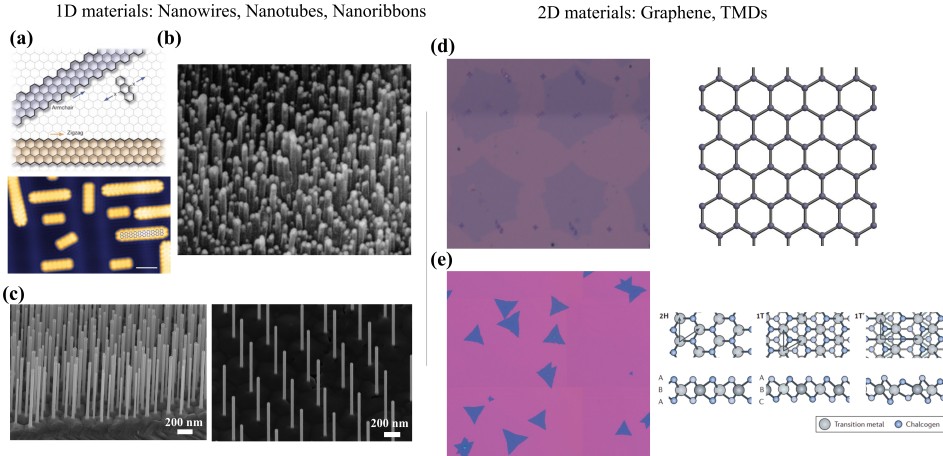

**Figure 2.** Zoology of nanostructures to be integrated with silicon photonic platforms to enhance device efficiency and lower the spatial footprint. (**a**) STM topography of graphene nanoribbons, which can either be realized by selecting the zigzag or armchair direction. Images reprinted from [78], copyright Springer Nature 2016, and from [79]. (**b**) Scanning electron micrograph of multi−walled carbon nanotubes. Image reprinted from [80], copyright Elsevier B.V. 2002. (**c**) Scanning electron micrographs of bottom−up grown forests of III−V semiconductor nanowires. (**d**) Optical micrograph of CVD grown graphene flakes (**left**) and pictorial representation of the chemical structure of graphene (**right**). (**e**) Optical micrograph (**left**) and chemical structure (**right**) of transition metal dichalcogenide flakes of $WS_2$. Image reprinted from [81], copyright Nature Springer 2017.

Notably, the integration of semiconducting nanostructure in photonic platforms enables the exploitation of their functional operation as optically active media which can be optimized by tuning material properties and device architectures, while avoiding negatively affecting the spatial footprint of PICs. Notably, 1D and 2D materials differ in terms of beneficial improvements and functionalities they can introduce, as will be shown in Sections 2.2 and 2.3, and researchers need to adapt the platform of choice depending on the specific functionality at need. One-dimensional nanostructures are preferred when small and localized optically active elements are needed; for instance, nanowire quantum dots are often integrated on silicon photonic platforms as localized single photon emitters. Additionally, the high aspect ratio and reduced spatial extension of semiconducting nanowires make them ideal to introduce asymmetries in waveguides to enable, e.g., light polarization control or to give rise to collective phenomena involving many semiconducting elements. On the other hand, 2D materials often feature better performance in terms of light modulation and absorption thanks to their lateral extension and favorable optical properties. They are often integrated in photonic platforms as electrically tunable modulators and electrically responsive photosensors. Interestingly, in the context of realizing complex devices for applications in industrial and technological fields requiring large scales and device repeatability, a great effort has been spent in developing techniques and methods to easily integrate semiconducting nanostructures on photonic waveguides and devices employing large-scale-compatible approaches. Representative approaches are reported in Figure 3, where the main techniques for the integration of 1D and 2D nanostructures on target substrates are reported.

It is worth mentioning that both the integration of 1D and 2D nanostructures give rise to relevant challenges, mainly coming from the positioning and orientation of nanostructures on the target photonic devices. For instance, the integration of nanowire light sources on photonic platforms requires precise positioning in order to ensure good coupling with waveguides, and even minimal displacements can induce loss in device per-

formance. Furthermore, the integration of a large number of nanowires can pose serious issues, for instance in terms of nanostructures' orientation and relative alignment. Two-dimensional materials pose similar challenges: in this case, orientation is less critical, but typically 2D nanostructures are fragile in terms of mechanical stress and can undergo damage during the transfer process, hampering device performance. For these reasons, the development of effective transfer techniques for both 1D and 2D nanostructures is of utmost importance for the development of effective integrated photonic devices boosted by semiconducting nanostructures.

In this context, the high aspect ratio of elongated 1D nanostructures led to the development of several techniques, including the following:

- Direct growth, involving growing the nanostructures directly on the target substrate using methods such as chemical vapor deposition (CVD) or molecular beam epitaxy (MBE) [82]. For instance, InAs/GaAs quantum dots have been grown on Si substrates [83] by MBE for the on-chip integration of lasers. This offers unparalleled control over nanowire position and orientation, leading to high-quality interfaces between the nanowires and the substrate. This method provides intriguing advantages, such as high material quality and the possibility to control the growth positions of the nanostructures. However, it is typically limited to specific substrate materials compatible with the nanowires growth conditions and can be challenging for achieving large-scale or complex nanowire patterns.

- Contact printing, a method involving physically pressing the 1D nanostructures grown on a source substrate against the target substrate. An example of application of such method is reported in [84], where the deposition of aligned Ge nanowires on a Si/SiO$_2$ substrate is demonstrated. Contact printing is a rather cost-effective method which has relevant advantages, including speed and ease of execution, allowing the transfer of a large number of nanostructures to a target substrate. However, its disadvantages include lack of fine control over nanostructures' positioning on the target and risk of physically damaging the sample due to mechanical stress. Overall, this approach is suitable for large-scale transfers but offers limited control over the precise positioning and alignment of individual nanowires.

- Fluidic assembly, leveraging the power of fluid flow or external forces like electric fields to guide nanowires suspended in a liquid to specific locations on the target substrate [85], as demonstrated with ZnSe nanowires deposited in μm-sized areas with dense packing of the order of $60 \times 10^3$ μm$^{-2}$ [86]. This approach allows for some control over the nanowire position and density within the fluid and enables the creation of more complex nanowire patterns compared to contact printing. However, achieving accurate alignment can be challenging, and careful control over the fluid flow and external forces is crucial to avoid harming the nanowires. Overall, fluidic assembly of 1D nanostructures provides notable advantages, such as the possibility to manipulate a large number of elements and compatibility with large-scale processing, but is limited in terms of fine control over the positioning of each individual nanostructure, and often relies on costly or difficult to manipulate liquids.

- Pick-and-place methods, utilizing micro-probes to physically manipulate and deposit individual nanostructures onto the target substrate [87]. For instance, Zadeh et al. [88] have demonstrated deterministic positioning of nanowire quantum dots employing a nanomanipulation setup composed of a tungsten tip mounted on a movable stage under a high resolution optical microscope. This technique's slow nature and labor intensity render it impractical for large-scale production. Despite this disadvantage, deterministic pick-and-place of individual nanostructures provides incomparable control over each nanostructure's positioning and orientation, providing a unique advantage in terms of device architecture control. For this reason, it is the golden standard for the realization of complex geometries for proof of concept and fundamental demonstrators.

- Transfer printing, involving employing an intermediate stamp, often made of a polymer, to transfer nanostructures grown on a donor substrate to a target substrate [89]. The work from Chang et al. [90], in which aligned ZnO nanowires have been transferred on a host substrate using a PDMS stamp, is illustrative of such method. The main advantages provided by this technique are greater control over nanowire positioning, cost efficiency and the possibility to deterministically place both individual nanostructures as well as arrays of nanostructures onto target substrates with relatively high precision. However, the main disadvantages of transfer printing are represented by potential contamination or residue from the stamp and limitations imposed by the adhesion properties of the stamp are factors to consider.

All the techniques reported here to transfer 1D nanostructures to target substrates are valid to integrate semiconducting nanomaterials in photonic platforms, but the specific method of choice needs to be assessed depending on the application and compatibility of the transfer process with other materials composing photonic devices. Indeed, if precise positioning and orientation of an individual nanostructure is targeted, direct growth—when applicable—or pick-and-place appear to be the best suited techniques to implement hybrid nanomaterial photonic device integration. On the other hand, if large-scale processing is needed, other techniques should be considered. For instance, contact printing is well suited for mechanically robust nanomaterials. If this is not the case, fluidic assembly or transfer printing—depending on the compatibility of the target substrate's material with liquid processing—should be considered.

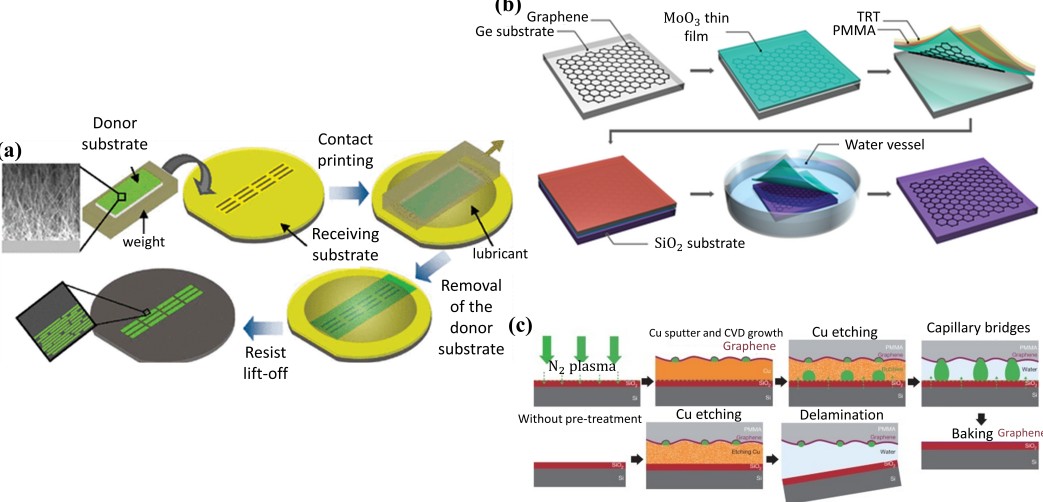

**Figure 3.** Most widely employed techniques for nanostructures deposition on host substrates applicable at the wafer scale. (**a**) Contact printing relies on the mechanical detachment of 1D nanostructures from a growth (donor) substrate on a host substrate. Image reprinted from [84], copyright ACS 2007. (**b**) $MoO_3$ assisted pick−up transfer technique of CVD grown graphene on a target substrate. Image reprinted from [91]. (**c**) Face−to−face transfer method for CVD graphene grown on Cu foils. Image reprinted from [92], copyright Springer Nature 2013.

On the other hand, different strategies have been developed for the integration of 2D nanomaterials in target substrates, including the following:

- Epitaxial growth transfer, where the 2D material is directly grown on a sacrificial substrate that shares a similar crystal structure, employed, for instance, for the growth of monolayer $MoS_2$ on silicon waveguides [93]. Subsequently, the sacrificial layer is selectively etched, enabling the 2D material to be transferred onto a target substrate with minimal lattice mismatch. Main advantages of epitaxial growth transfer are the high quality of the resulting materials and the possibility to selectively grow nanostructures on specific locations thanks to assisted growth techniques. However, this method is limited by the growth conditions of the specific material under consideration, which

may not be compatible with the target substrate (e.g., growth temperature may be too high, resulting in damage to the target substrate). Overall, this method offers excellent control over the crystallographic orientation and minimizes interface defects, making it ideal for specific device applications. However, it is limited to compatible material combinations and can be more complex to implement compared to other techniques.

- Wet transfer, involving submerging the growth substrate with the 2D material in a sacrificial layer, typically a polymer film [92,94]. The sacrificial layer is then lifted onto the desired target substrate, followed by dissolving the sacrificial material using a solvent, leaving the 2D material deposited on the target. Exploiting this method, large-scale transfer of CVD-grown graphene using polyvinyl alcohol polymer foils was demonstrated without relevant losses in material quality in terms of residual doping [95]. This method has the major advantage of being cost-effective and straight-forward, but simultaneously suffers from several disadvantages: indeed, it can introduce surface contamination and limit the choice of solvents compatible with both the sacrificial layer and the 2D material. Additionally, for some applications the use of solutions processes involving solvents may hamper device functionalities and induce materials degradation.

- Dry transfer, aiming to eliminate the use of liquids and minimizing the risk of contamination. This approach involves using a polymer stamp to pick up the 2D material from the growth substrate and subsequently transfer it to the target [96]. This approach allowed for the successful transfer of epitaxial graphene grown on SiC to $SiO_2$, GaN and $Al_2O_3$ target substrates [97]. This method shines for several advantages, including cleanliness and the possibility to deterministically position 2D materials onto target substrates thanks to proper alignment of the polymeric stamp. On the other hand, dry transfer also has several disadvantages. For instance, it is a time-consuming technique and is hardly compatible with large-scale processing, making it viable only at laboratory scale.

- Electrochemical delamination, utilizing an electric field applied through an electrolyte solution to selectively etch the sacrificial layer, releasing the 2D material that can then be transferred to the target substrate [98,99]. Notably, this technique has been employed to transfer an ordered array of deterministically positioned CVD-grown graphene flakes to enable large-scale device fabrication [100]. This method offers precise control over the transfer process and minimal surface contamination. Indeed, electrochemical delamination features the promising advantage of greatly preserving material quality, resulting in optimal values of relevant parameters, e.g., electrical mobility. However, it requires careful control of the electrical parameters and is limited to specific growth and target substrate combinations. Additionally, electrochemical delamination shares the same disadvantage as wet transfer, namely, the presence of solvents and liquids which may not be compatible with specific material platforms.

Similarly to the 1D case, the choice of a specific technique for transferring 2D materials to target substrates for the realization of hybrid integrated photonic devices depends on specific implementation needs. For instance, laboratory-scale experiments in which high material quality is needed to realize proof-of-concept demonstrators would strongly benefit from the employment of epitaxial growth transfer. On the other hand, wet transfer and dry transfer are more cost-effective and large-scale compatible techniques. Furthermore, electrochemical delamination is optimal for large-scale applications in which the use of solvents and electrochemical reactions do not significantly damage other device components.

Indeed, all the techniques described here—both for 1D and 2D nanostructures—present advantages and disadvantages for their application, and which one is to be employed typically depends on the specific application in development, as well as the degree of control over the positioning and orientation of the nanostructure required in order to ensure the effective operation of the integrated photonic device to be realized.

### 2.2. One-dimensional Nanostructures Integration in Integrated Photonic Platforms

The integration of 1D nanostructures in integrated silicon photonic platforms has enabled the increase of performance and the introduction of novel functionalities provided by on-chip elements. Indeed, high-aspect-ratio nanostructures such as nanowires can allow for the realization of complex high-quality heterostructures [60], allowing, e.g., for the realization of single photon quantum emitters/detectors, a technology otherwise hardly accessible. Additionally, the low spatial footprint of 1D nanostructures allows for their integration in PICs with increased flexibility, thus being able to exploit the combined effect of many nanostructures, e.g., to provide periodic perturbations or exploit avalanche effects. Figure 4 reports examples of photonic devices combining silicon platforms with semiconducting 1D nanostructures.

In this context, a novel polarization control device has been proposed [101], allowing for the achievement of full polarization rotation within a reduced spatial scale (<20 μm) and with low losses (<2 dB). This is achieved by introducing a set of semiconducting nanowires on a Si waveguide in order to provide a periodic perturbation leading to the controlled rotation of the polarization. Additionally, the number and distance between nanowires has been observed to have a profound effect on the rotation effect, leading to the possibility to tune device operation in order to implement arbitrary polarization rotations, an essential feature needed for the operation of photonic circuits [13,102]. Furthermore, other requirements of photonic devices have been fulfilled thanks to the integration of high-aspect-ratio nanostructures on silicon photonic platforms, including mode filtering achieved with the integration of graphene nanoribbons and featuring $TE_1$ mode pass filtering with a $TE_1$-to-$TE_0$ extinction ratio of 9.19 dB [103] and wavelength filtering employing silicon-on-insulator nanowires [104], enabling wavelength conversion up to 160 Gb/s with minimal power loss (<3 dB). Additionally, the favorable optical properties of semiconducting nanostructures in terms of light emission and detection have been combined with PICs to provide on-chip emitting and detecting functionalities. For instance, carbon nanotubes were integrated with Si waveguides at 1.3 μm wavelengths [105], successfully demonstrating emission and detection functionalities. Operation at 1.5 μm with carbon nanotubes has also been achieved [106] and a carbon nanotube-based photodetector with 48 GHz bandwidth featuring responsivity up to 73.62 mA/W has been demonstrated [107]. This class of nanostructures has also been employed for the implementation of logic operation with cascading nanotubes-based photodetectors [108]. On the other hand, semiconducting nanowires allow for more complex heterostructures and doping control compared to their hollow counterparts, owing to bottom-up growing techniques and strain relaxation [60]. For this reason, material properties are more easily controlled for this class of nanostructures, for instance, allowing the realization of multiple p/n junction in series on the same nanowire. This leads to novel possibilities: for instance, nanowires featuring repeated p-i-n junctions were integrated in Si waveguides to enable efficient photodetection [109]. Furthermore, owing to the subwavelength emission capabilities of semiconducting nanowires, their integration on silicon photonic platforms to enable on-chip light emission has been investigated, both in classical [110,111] and quantum [112–114] regimes.

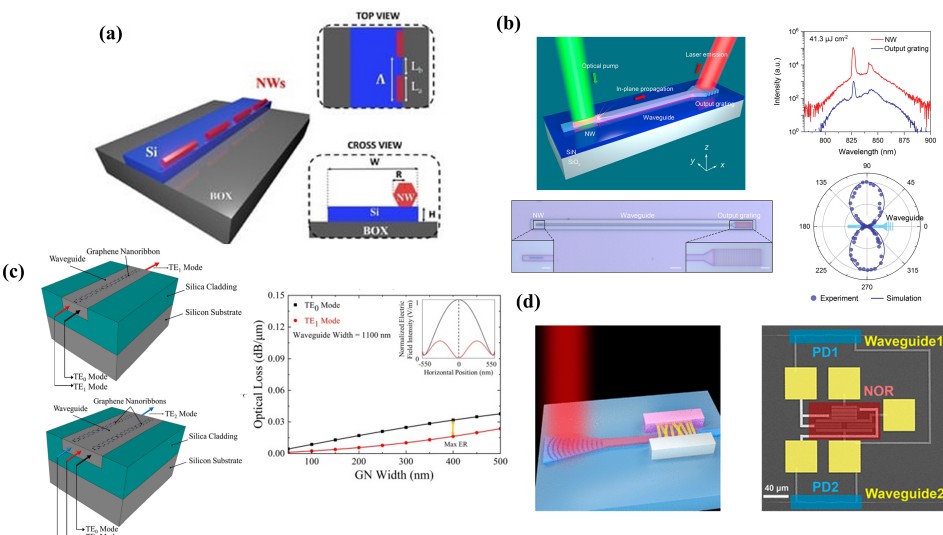

**Figure 4.** Integrated photonics platforms employing 1D nanostructures. (**a**) Polarization control devices can be realized by combining semiconducting nanowires with silicon waveguides. Image reprinted from [101]. (**b**) Single nanowire−based light sources embedded on−chip for light generation in PICs. Image reprinted from [111], copyright ACS 2020. (**c**) PICs integrated with graphene nanoribbons to deploy mode filtering in silicon waveguides. Image reprinted from [103], copyright ACS 2022. (**d**) Integration of carbon nanotubes photodetectors in PICs to achieve on−chip light detection. Image reprinted from [108].

### 2.3. Two-dimensional Materials Integration in Integrated Photonic Platforms

Two-dimensional materials, similarly to their one-dimensional counterparts, present a vast zoology of different materials featuring different physical properties, spanning from gap-less monolayer graphene to the TMDs featuring tunable direct bandgap. Generally speaking, layered materials present specific challenges with respect to sample manipulation, but the positioning of 2D flakes on silicon integrated photonic circuits is less critical compared to elongated nanostructures. Furthermore, the huge experimental effort towards the development of optoelectronic devices based on graphene and TMDs has inevitably cross-fertilized the employment of these materials as components in PICs [115].

In this context, TMDs provide an ideal material platform to be employed in integrated photonic devices, owing to their flexibility in terms of material properties [116] and access to reversible tuning of their electrical properties by exploiting ionic liquid gating—a non conventional gating technique which has been proven to outperform conventional gating techniques [117,118]. Figure 5 shows representative examples of the integration of TMDs on silicon photonic platforms. The integration of TMDs in integrated photonic platforms has been investigated in order to develop photonic circuitry exploiting the aforementioned promising features of this material class [119], enabling the development of essential photonic devices such as Mach–Zehnder interferometers [120]. Additionally, the possibility to tune the effective refractive index of TMDs by means of bandgap engineering [116] or by exploiting electrostatic doping via ionic liquid gating [121,122] enabled the realization of photonic modulators [123,124]. For instance, Joshi et al. [125] have realized electro-optical modulators with different TMDs integrated in silicon nitride waveguides, investigating their performance in terms of modulation strength and finding that in their device geometry $WS_2$ resulted in the strongest modulation (19.88%), followed by $WSe_2$ (9.48%), $MoS_2$ (3.72%) and $MoSe_2$ (2.95%). Another essential functionality which needs to be available on photonic platforms and which cannot be fulfilled by silicon owing to its indirect bandgap—i.e., light generation—has been addressed exploiting TMDs, and specifically employing electrically tunable p-n junctions [126], which can act both as light emitters and detectors. Furthermore, additional coupling mechanisms between TMDs and photonic elements have been investigated, such as the interconnection between excitons in

TMDs—observable at room temperature—and plasmonic modes [127,128]. In this context, 2D materials hold great promises to boost a fast-developing field within photonics in general, i.e., plasmonics [129], and in silicon photonics specifically [130]. Indeed, the possibility to exploit the propagation of surface plasmon polaritons at the interface between a dielectric and an electrically charged region enables confinement in the sub-wavelength regime, allowing for device miniaturization [129]. However, these devices are typically realized using metals to host the free charges to couple with the electromagnetic wave, resulting in high losses due to heating effects [131]. For this reason, the development of novel material combinations to enable the realization of effective plasmonic devices and overcoming the limitations coming from the use of metals is rapidly proceeding forward [132], and 2D materials stand as promising candidates in the race for the realization of plasmonic waveguides.

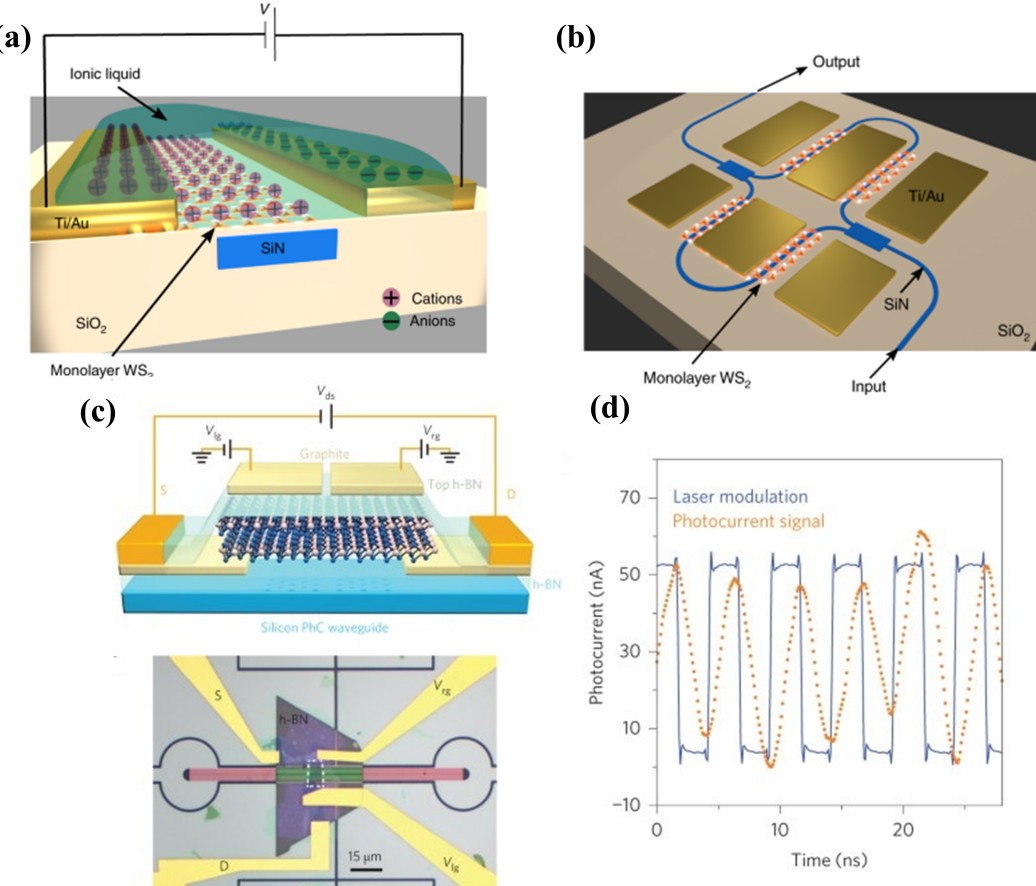

**Figure 5.** Examples of integrated photonic devices combining silicon and TMDs. (**a**) Integration of $WS_2$ in PICs exploiting ionic liquid gating to tune the effective refractive index of the material. Image reprinted from [120], copyright Nature Springer 2020. (**b**) The same platform can integrate Mach−Zehnder interferometers to probe the phase shift of guided mode, to investigate its dependence on the combination between light polarization and $WS_2$ electrostatic doping. Image reprinted from [120], copyright Nature Springer 2020. (**c**) $MoTe_2$ electrically tunable p−n junction integrated with silicon photonics waveguides to achieve on−chip light generation and detection. Image reprinted from [126], copyright Nature Springer 2017. (**d**) Response of the $MoTe_2$ light detector. Image reprinted from [126], copyright Nature Springer 2017.

Together with TMDs, another well-known 2D material in the field is graphene. Indeed, this material has been extensively studied since its discovery, leading to the development of great advances regarding its electrical and optical properties. For instance, high-mobility graphene is readily available both in the laboratory as well as in larger scales, and it can

be processed by means of scalable techniques [133]. Figure 6 reports some examples of photonic devices integrating graphene with silicon photonic platforms.

In combination with high mobility [134], the semi-metallic nature of monolayer graphene makes it an ideal material for high speed photodetectors. Indeed, this functionality has been readily added on integrated photonic platforms [135], allowing them to reach good performance in terms of photoresponse and signal processing speeds. For instance, Schiue et al. [136] reported the realization of a graphene-based photodetector integrated on a SiO$_2$ waveguide featuring a responsivity of 0.36 A/W with a cutoff frequency at 3 dB of 42 GHz, while Schall et al. [137] reported high-speed photodetectors based on CVD graphene working at 1.5 µm with data rates up to 50 GBit/s and featuring a bandwidth of 41 GHz. An increase of performance has been investigated for the realization of waveguides as well, exploiting the capability of graphene to couple with plasmonic resonances and to realize plasmonic photonic waveguides [138,139]. Despite graphene being limited in terms of light generation due to the absence of a bandgap, its favorable electrical transport properties allowed for the integration on silicon waveguides, enabling the realization of effective optical modulators [140–143], reaching modulation efficiency up to 1 dB/V and data rate up to 20 Gbps [144] and switches [145]. Additionally, several proposals employed graphene to realize polarization control devices with fixed polarization filtering [146], as well as tunable filters [147,148].

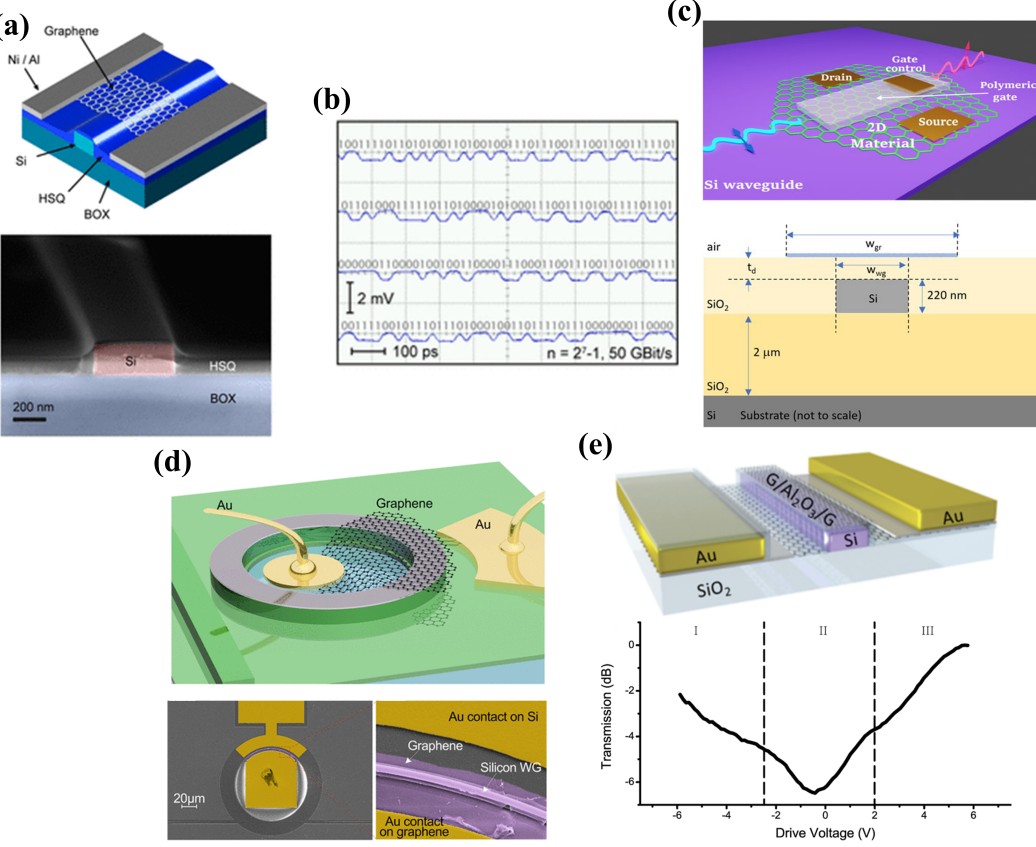

**Figure 6.** Photonics devices realized by combining graphene with silicon integrated platforms. (**a**) Integration of graphene photodetectors on silicon photonics waveguides. Image reprinted from [137]. (**b**) Fast response features by graphene allows for signal detection up to 50 Gbit/s. Image reprinted from [137]. (**c**) Tunable polarization control enabled by graphene integration in PICs. Image reprinted from [148]. (**d**) The integration of graphene on silicon photonic devices enables high efficiency electro−optical modulation. Image reprinted from [141] with permission from ACS. (**e**) Graphene−enabled optical modulation on silicon photonic platform achieved by exploiting a double−layer graphene p−oxide−n junction. Image reprinted from [140] with permission from ACS.

## 3. Conclusions

This review explored the significant advancements in merging integrated silicon photonics with semiconducting nanostructures. It provided insights into the principal methods employed for this integration, enabling the realization of enhanced photonic platforms that leverage the advantages of nanostructures. The review further highlighted the key achievements made possible by this synergistic approach, demonstrating how nanomaterials can not only significantly enhance device performance but also overcome inherent limitations of conventional technologies, paving the way for the introduction of novel device functionalities.

The central role of semiconducting nanostructures in the development of next-generation photonic platforms appears undeniable. However, further advancements in this field will require effort on two fronts. Firstly, the continuous development of scalable fabrication processes is crucial for cost-effective and high-throughput production of integrated photonic platforms, laying the foundation for widespread adoption of this technology. Secondly, ongoing refinement of nanostructure integration techniques will unlock the full potential of these materials to further enhance device performance and functionality, ultimately leading to the realization of advanced photonic devices with unprecedented capabilities. By pursuing these advancements the field of integrated silicon photonics empowered by semiconducting nanostructures has the potential to revolutionize various sectors, including telecommunications, healthcare and quantum computing.

**Author Contributions:** All authors contributed to this work: writing—original draft preparation, D.P.; writing—review and editing, all authors. All authors have read and agreed to the published version of the manuscript.

**Funding:** This work was funded by INFN through the CSN5 QUANTEP project.

**Acknowledgments:** F.R. and L.M. acknowledge the National Recovery and Resilience Plan (NRRP), Mission 04 Component 2 Investment 1.5–NextGenerationEU, Call for tender n. 3277 dated 30/12/2021. Award Number: 0001052 dated 23/06/2022. V.B. (Vittorio Bellani), G.D.G., A.F., R.G., C.L., E.P., P.P. (Paolo Piergentili), V.R., C.R., A.S. (Andrea Salamon), and F.S. (Franco Spinella) acknowledge the support of the PNRR MUR project PE0000023-NQSTI (Italy).

**Conflicts of Interest:** The authors declare no conflicts of interest.

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
