# Peer review of "Hybrid Integrated Silicon Photonics Based on Nanomaterials"

_photonics, doi:10.3390/photonics11050418_

Round 1
Reviewer 1 Report
Comments and Suggestions for Authors
D. Prete et al. presented a review of the advancements in the field of hybrid nanomaterials for applications in integrated photonics.
The key advantage of the extensive review apart from presenting a wide range of examples is the extended discussion taking into account both material properties and the fabrication technologies required compatible with well-established protocols of silicon photonics.
The manuscript might be of high interest to a wide range of readers, however, I suggest several improvements to make the work easier to read for a broader audience.
· In my opinion, the title is not adequate for the body of the manuscript. The work discusses the application of several types of materials in integrated silicon photonics, thus I suggest specifying the title according to the topic of the manuscript.
· The authors cite several review papers on the undertaken topic. I suggest adding a short section describing what differentiates the presented manuscript from the available literature.
· The review is focused on pointing out the most prominent examples showing progress in the field, however, I strongly recommend sparing some space for considerations regarding the advantages and disadvantages of the discussed approaches.
· In lines 253-255 the authors state: “Furthermore, additional coupling mechanisms between TMDs and photonic elements have been investigated, such as the interconnection between excitons in TMDs — observable at room temperature — and plasmonic modes” This effect is mentioned once without further elaboration. In my opinion, the discussion on the impact of plasmonic nanostructures on the development of photonics devices might enrich the presented manuscript on the other class of materials being evaluated for the PICs applications. The authors can find several reviews on this topic (e.g.
1. P. Sun, et al Photonics 2021, 8, 482.
2. W. Heni, et al ACS Photonics 2017, 4, 1576−1590
3. F. J Rodríguez-Fortuño, et al J. Opt. 2016 18 123001)
4. S. Papaioannou, et al. "Merging Plasmonics and Silicon Photonics towards Greener and Faster “Network-on-Chip” Solutions for Data Centers and High-Performance Computing Systems." (Ed: K. Y. Kim), IntechOpen, London, UK 2012
· The authors should name the most critical parameters of the photonic devices that might be improved by the integration with the nanostructures instead of naming them in general as the device performance.
· Figures presented in the manuscript are of low resolution and have small labels which make them unreadable.
Reviewer 2 Report
Comments and Suggestions for Authors
The review on "Hybrid Integrated Photonic Nanomaterials" presents a promising exploration of the integration of nanoscale materials into silicon photonic devices. However, to enhance the overall impact and rigor of the review, a more thorough revision is necessary. The following feedback provides specific areas that require attention, including the need for a more detailed analysis of integration methods, a discussion on challenges and solutions, and a deeper exploration of quantitative data to support the claims made. Addressing these aspects will significantly strengthen the review and contribute to its acceptance in the academic community.
1 The text mentions various techniques for integrating nanomaterials (e.g., direct growth, contact printing, transfer printing) but lacks specific examples or comparisons of their effectiveness. Providing a more in-depth analysis or citing studies that compare these techniques could strengthen the review.
2 In Section 2.1, where integration techniques are discussed, consider providing a more detailed analysis of each technique's advantages and disadvantages. This will help readers understand the practical implications and trade-offs associated with each method.
3 The text could benefit from a discussion on the challenges encountered in integrating nanomaterials into silicon photonic devices and the proposed or potential solutions to these challenges.
4 While the review discusses 1D and 2D nanostructures separately, a direct comparison or discussion on when to prefer one over the other in specific applications would add depth to the analysis.
5 The review mentions that nanoscale materials enhance device performance, but it would be more impactful if supported by quantitative data or comparisons with traditional silicon photonic devices.
Reviewer 3 Report
Comments and Suggestions for Authors
This work offers a short review on the latest endeavors in a hybrid platform silicon/nanomaterials whose primary aim is to enchance the performance of photonic integrated devices based on Silicon.
The results presented display the latest advances in this kind of hybrid platform. The references are appropiate and adequate.
However, I would like to suggest that the title should be slightly modified because the main platform used is based on Silicon, in which nanomaterials are incoporated. That is, the word: "Silicon" would have to appear in the ttile.
Alternatively, if the title is not changed then other platforms would have to be considered, such as: Semiconductors II-V (AsGa, InP), SiN, SiC, InP/SOI, Diamond on Insulator, LN, Glass and so on, and to show results of hybrid integration in these platforms with nanomaterials.
On the other hand, the size of the Figures must be enlarged. Many details are very difficult to appreciate. It is true that they can be enlarged with a digital zoom, but in my opinion Figures should look well in the standard space of size A4, particularly in a review where these Figures are essential to see and understand the results.
Comments on the Quality of English Language
The quality of English is enough, although minor corrections have to be made, as for example in line 120: "nanostructruses", and so on.
Round 2
Reviewer 1 Report
Comments and Suggestions for Authors
The authors adequately responded to my suggestion. Especially the section devoted to the fabrication methods has been extended by the discussion on the pros and cons of the mentioned techniques which differentiate the manuscript from other similar works. In my opinion, the manuscript in its present form can be considered for publication in the Photonics journal.
Reviewer 2 Report
Comments and Suggestions for Authors
The authors have addressed the reviewer's comments. So, this paper should be published in Photonics.